# Examining women's choice between home and institutional births: Insights from the Salud Mesoamérica Initiative (SMI)

Lucía Bartolomeu[1]*, Natalia Basualdo[1], Federico Rolón[1], Adolfo Rubinstein[1],
Camila Volij[1], Matt Simon[2], Brittany Hagedorn[3], Diego Ríos-Zertuche[4],
Santiago Esteban[4], Adrián Santoro[1]

1 Center for Implementation and Innovation in Health Policies, Institute for Clinical Effectiveness and
Health Policy, Buenos Aires, Argentina, 2 Primary Health Care, Global Development Division, Gates
Foundation, Seattle, Washington, United States of America, 3 Institute for Disease Modeling, Global
Health Division, Gates Foundation, Seattle, Washington, United States of America, 4 Health, Nutrition and
Population Division, Inter-American Development Bank, Washington, United States of America

* lbartolomeu@iecs.org.ar

## Abstract

This study examines women's childbirth choices between home and institutional
settings in the context of the Salud Mesoamérica Initiative, a results-based financing
program focused on disadvantaged populations. This study characterizes the child-
birth experience using the World Health Organization's Positive Childbirth Experience
framework, identifies factors influencing women's decisions for home versus institu-
tional births, and describes reasons for transitions between these settings, especially
for women with multiple pregnancies. A retrospective observational study was carried
out using secondary data from household surveys carried out in Guatemala, Chiapas
(Mexico), Nicaragua, and Honduras. A two-stage random sample of women (15–49
years) who had a live birth within five years preceding the surveys was analyzed.
Quality of institutional care and reasons for home birth were categorized into specific
domains and data across three follow-up points were analyzed using design-adjusted
chi-squared tests (Rao-Scott correction). Finally, for the second follow-up, a sub-
analysis explored setting transitions for women with multiple pregnancies. The main
results indicate that Guatemala had the highest rate of home births (83.8%), while insti-
tutional deliveries were more common in Honduras and Nicaragua. Over time, institu-
tional births increased across the board, notably in Honduras, where they surged from
72.3% to 92.7%. The primary factors influencing birth setting choices were accessibility
and cultural preferences, with the latter being particularly significant in Guatemala and
Chiapas. The results underscore the critical need for culturally sensitive strategies to
boost institutional birth rates. While enhanced accessibility has contributed to more
institutional deliveries, deep-seated cultural preferences remain a considerable hurdle.
Interventions that respect local beliefs while simultaneously improving the quality of
care in institutional settings are essential to ensure safe childbirth for all women.

**Data availability statement:** The original data and R scripts required to reproduce the statistical analysis and visualizations presented in this study are openly available in the following GitHub repository: https://github.com/LuciaBart/SMIchildbirth/.

**Funding:** This manuscript is part of the project "In-Depth Analysis of Salud Mesoamerica Initiative Data for Primary Healthcare Improvement," funded by the Gates Foundation (Grant INV070172, 2024).

**Competing interests:** The authors have declared that no competing interests exist.

## Introduction

Ensuring respectful maternity care and a positive childbirth experience is a critical public health priority in the region, where maternal and neonatal health outcomes remain a challenge due to persistent inequalities in access and quality of care [1,2]. Despite significant progress in reducing maternal and perinatal mortality, disparities in obstetric care—particularly among indigenous and low-income populations—continue to affect health equity [3]. High rates of obstetric violence, unnecessary medical interventions, and limited access to culturally appropriate care further highlight the need to strengthen evidence-based, woman-centered childbirth practices [4].

Global efforts are made to reduce these gaps. The World Health Organization's (WHO) framework for a Positive Childbirth Experience underscores the importance of respectful, dignified, and high-quality maternal care for improving both immediate and long-term health outcomes, emphasizing its importance as a key result for all women in labor. In this context, a positive experience in childbirth surpasses a woman's pre-existing personal and sociocultural expectations, including delivering a healthy baby in a clinically and psychologically secure environment, with ongoing practical and emotional support from chosen birth companions and compassionate, skilled clinical personnel. The basis of this approach is the understanding that most women prefer natural labor and delivery and desire to feel empowered and involved in the decision-making process, even when medical interventions are deemed necessary or desired [5].

The Salud Mesoamerica Initiative (SMI) was a result-based financing program primarily to improve mother and child health outcomes in Mexico and Central America, and intended to be a transformative framework for advancing reproductive health across the region. This public-private partnership addressed health inequities by targeting countries' most vulnerable populations, particularly women and children. The initiative emphasized strengthening healthcare systems, improving access to quality maternal and reproductive health services, and reducing barriers to contraception and family planning [6].

SMI was implemented across eight countries (Belize, Costa Rica, El Salvador, Guatemala, Honduras, Mexico -specifically the state of Chiapas-, Nicaragua, and Panama). Within each country, the initiative focused on districts representing the poorest 20% of the population.

According to studying protocols from SMI Initiative, the rationale for prioritizing institutionalized childbirth stems from the critical health disparities and persistently high maternal and neonatal mortality rates in the targeted regions. In 2007 maternal mortality remained as high as 136 per 100,000 live births in Guatemala. Disproportionate burden exists between rural and indigenous populations: in Chiapas, for instance, institutional births among indigenous women were as low as 42.4%, compared to the 69.7% state average before the beginning of the Initiative. This lack of skilled birth attendance is a primary driver of preventable deaths; obstetric hemorrhage—a condition requiring immediate clinical intervention—accounts for 33% of maternal deaths in Chiapas, while asphyxia and trauma cause 25% of neonatal fatalities. Furthermore, in countries like Nicaragua, the risk of neonatal death is five

times higher for children born to adolescent mothers, a group that represents 20% of maternal mortality. By encouraging institutionalized birth, the SMI aimed to bridge the gap between vulnerable populations and the life-saving infrastructure—such as emergency obstetric care and skilled human resources—necessary to manage these high-risk complications that home settings are unequipped to handle.

Between 2012 and 2021, a baseline and two or three follow-up surveys were carried out depending on the country. For this paper, we refer to these as baseline, second follow-up,and third follow-up. Secondary analysis of this data is an opportunity to get more information about the childbirth experience in the most vulnerable population.

This study aimed to describe the childbirth experience of the mothers according to the new WHO's positive childbirth experience framework, to analyze the choicein the place of delivery in each country, and to understand the underlying reasons for changes in birth delivery setting among women with multiple pregnancies who transitioned between institutional births and home births, in either direction.

## Materials and methods

### Data

This is a retrospective observational study based on secondary analysis of data collected within the framework of the Salud Mesoamérica Initiative. According to the Initiative, countries have to work within the poorest 20% of their populations, selected based on Poverty Incidence Data. Country operations were directed toward geographical areas or populations classified as extremely poor and/or indigenous, based on national definitions. In this study four selected countries were analyzed, as defined by the Initiative: State of Chiapas, Nicaragua, Honduras and Guatemala. For each of them, according to the available surveys performed in each location, two or three points in time were analyzed: for the baseline (2012–2014 for all countries), Second (2015–2017 for Nicaragua, Honduras and Guatemala and 2015–2018 for Chiapas) and Third Follow-up (FUP) (2018–2021, only available for Nicaragua and Honduras).

During the SMI Initiative, a household survey was performed for each country. The surveys employed a two-stage random sampling method to efficiently collect data while ensuring the results accurately reflect the overall population. First, according to the Program objectives, target areas were defined in each country to provide estimates of the coverage of key health interventions and indicators among the lowest wealth quintile of the population. In the first stage, from those target areas, primary sampling units (PSUs) were randomly selected. In the second stage, a sample of households were drawn from each selected PSU. All women aged 15–49 years who are residents of the household are eligible to be interviewed. This approach balanced efficiency with statistical precision. SMI measurement methods are characterized based on the methodology described in the documents and reports protocols from the Initiative [6,7].

Data from the household surveys conducted in the four countries and all available follow-ups (Baseline, Second FUP, and Third FUP for Nicaragua and Honduras; Baseline and Second FUP for Guatemala and Chiapas) were analyzed in this study. Women who had a live birth in the 5 years preceding the surveys were included. As cesarean sections could only occur at institutions, those records coming from women with c-sectionwere excluded from the analysis. This criterion was applied to ensure the inclusion of women who could choosefor a home birth or an institutional delivery. The analyzed questions focused on labor and delivery for each birth occurring within the five years preceding each follow-up.

For Nicaragua and Honduras, data were available from the baseline (BL), second follow-up, and third follow-up. In contrast, only BL and second follow-up data were available for Guatemala and Chiapas. The data collection phase of the SMI Initiative ended by 2021. For this study, the authors accessed public and anonymized finished datasets on April 1st., 2024. Obstetric complications data were not available for home-birth setting so they are excluded from the analysis.

### Domain analysis construction

In the household survey developed by SMI Initiative, different questions were performed depending on whether women had an institutional or a home birth.

Available data from women who had an institutional birth were related to the quality of care. The questions aligned with the WHO framework for a positive childbirth experience were selected. For women having a home birth available data was based onthe reasons for their home-birth decision. Both sets of variables from each birth setting were categorized into several domains for analysis, following the WHO's positive childbirth experience framework (World Health Organization 2018). The domains associated with both quality of care in institutional births and factors influencing home birth choices are presented in Tables 1 and 2. Each domain consists of a series of corresponding household survey questions, detailed in the supporting information (S1 and S2 Tables).

### Place of delivery

To analyze differences in the distribution of birth type (home versus institutional delivery) across the follow-ups (baseline, second follow-up, third follow-up) within each country, a proportion test strategy was implemented. Specifically, design-adjusted chi-squared tests were performed using R statistical software (version 4.4.2). Given the survey design, which involved multi-stage sampling, primary sampling units and sample weights were incorporated into the inferential analysis to obtain valid and population-representative results. A significance level of $p < 0.05$ was used for all.

### Construction of the quality scores for women who had an institutional delivery

As each domain contains several questions, the whole domain was considered positive if at least one of its corresponding questions received an affirmative response. Thus, four dichotomized domains (1,0) were analyzed for each institutional birth.

To assess differences in the proportions of institutional childbirth quality indicators over time, comparisons were made between three follow-up points (Baseline, Second FUP, and Third FUP), stratified by country. Since the data came from surveys with a complex design, the R software was used to adjust the analyses for sampling weights, stratification, and primary sampling units. Proportions and their 95% confidence intervals were estimated for each domain. For group comparisons, chi-square tests adjusted for design with Rao-Scott correction were applied.

Also, a quality score for each birth was obtained from the sum of the four domains for quality analysis. Depending on the number of positive domains achieved, each birth had a global quality score of 0, 1, 2, 3, or 4 (0 being the lowest and 4 the highest).

**Table 1. Domains adapted from the WHO positive intrapartum experience for institutional births.**

| Institutional birth: Domains for variables related to quality of care |
| --- |
| 1. Respectful treatment |
| 2. Effective communication |
| 3. Accompaniment |
| 4. Recommended practices |

**Table 2. Domains in home births.**

| Home birth: Domains for variables related to reasons for home birth |
| --- |
| 1. Knowledge |
| 2. Accessibility |
| 3. Infrastructure |
| 4. Respectful treatment and Effective communication |
| 5. Family |
| 6. Cultural preferences |

**Reasons for home birth choice over time for women who had a home delivery**

The reasons for home birth were grouped into six domains: (Table 2 and S2 Table) according to previous publications of the SMI. As each domain contains several questions, the whole domain was considered positive if at least one of its corresponding questions received an affirmative response. The percentage of each reason domain was calculated for each country and follow-up.

**Analysis of women who shifted birth delivery setting**

At the second FUP, where we can ascertain the reproductive history of each woman, we analyzed the possible reasons behind the change of the birth setting in 207 women who switched the birth settings for their last two births.

Women who changed their birth delivery setting between their two most recent deliveries were classified into two categories: those whose first child was born in a health facility and the second at home (institutional to home birth) and those whose first child was born at home and the second in a health facility (home to institutional birth). Within each category, for each woman, we analyzed the two birth scenarios: reasons for choosing home birth in case of home birth, and the health facilities' perceived quality of care in case of institutional deliveries, based on available data for each birth-setting. Although the questionnaire does not ask exactly why the woman decided to change the type of delivery, a description of both birth scenarios was made for each woman to assess whether there was a relationship between the recorded change in the survey and the quality of care and/or the reasons for choosing a home birth.

**Ethics statement**

The study is based on a secondary data analysis generated by the Salud Mesoamerica Initiative from baseline and follow-up measurements. At the time of data collection written informed consent was obtained from all participants. The study received approval from the institutional review board (IRB) from the University of Washington, partnering data collection agencies, and the Ministry of Health in each country. For the study presented in this article, the authors accessed public and anonymized datasets on April 1st., 2024.

The original data and R scripts required to reproduce the statistical analysis and visualizations presented in this study are openly available in the following GitHub repository: https://github.com/LuciaBart/SMIchildbirth/.

## Results

Two different analyses were performed in this study. Initially, a brief description of birth settings distribution was done. Then, according to the birth-setting available data, a descriptive analysis of the quality of care at institutions and the reasons for home birth were performed. Secondly, women who shifted their birth setting specifically for the second FUP were analyzed.

A total of 29172 live births were analyzed from the 3 follow-ups. The flow chart is shown in Fig 1.

To evaluate women's choice of institutional birth, cesarean sections were excluded since they can only occur at institutions. C-section rate was: 25% in Baseline, 27% in Second-FUP and 25% in Third FUP.

**Descriptive analysis of the place of delivery**

Data collected from Guatemala, Chiapas, Honduras, and Nicaragua reveal notable variations in birth setting distributions. For each country and each follow-up, the weight percentage of home birth and institutional delivery was calculated, covering births within the five years preceding each survey, as shown in Fig 2.

At baseline, home births were predominant in Guatemala and Chiapas, reaching 83.8% (3910 births) and 66.9% (3562 births), respectively. In contrast, Honduras and Nicaragua reported significantly lower proportions of home births (27.7% and 15.1% representing 623 and 235 births, respectively), with most deliveries occurring in health facilities.

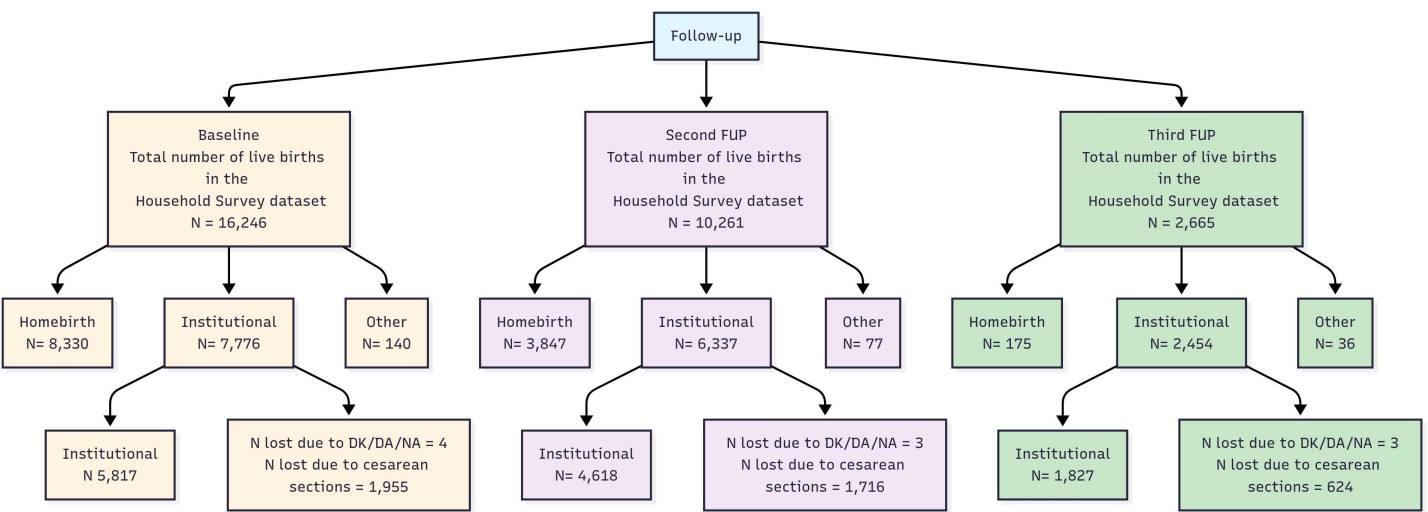

**Fig 1. Flowchart of the household survey registries analysed matched by initiative stage (baseline, second follow-up and third follow up).** Numbers indicate the total amount of live births in each category. DK/DA/NA: don´t know, didn't answer, not available.

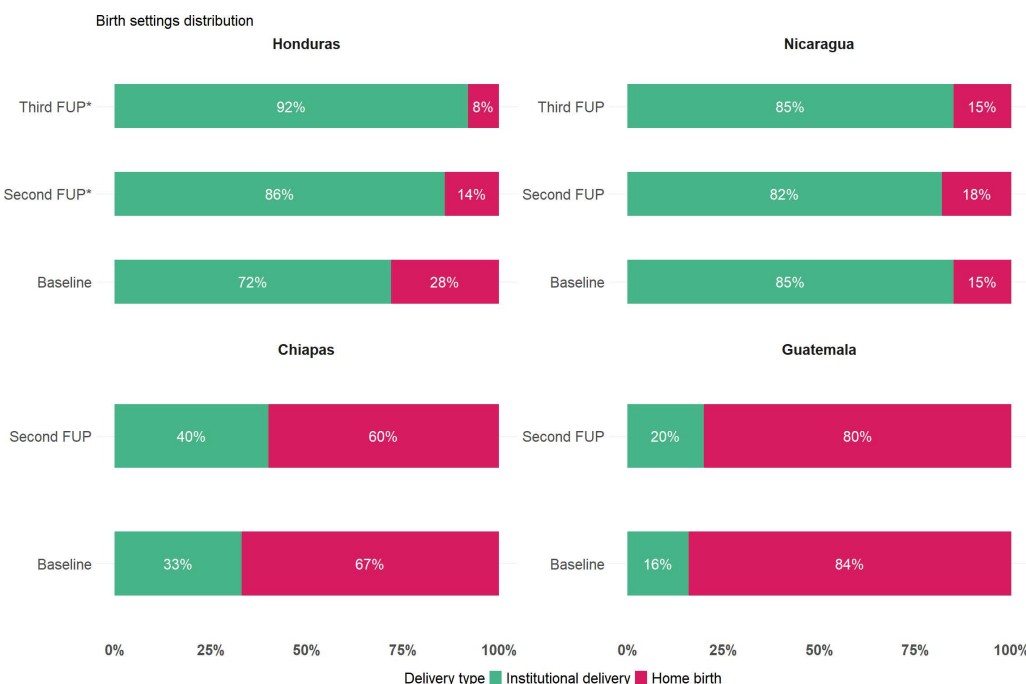

**Fig 2. Distribution of birth settings (home birth and institutional delivery) presented as weight percentages(%).** Data are shown for each country and follow-up period, covering births within the five years preceding each survey. * significant difference compared to Baseline follow-up (p-value<0.05).

The data indicate a general trend toward increased institutional deliveries along the study period, with this shift being particularly significant in Honduras, where institutional births rose from 72.3% at baseline to 92.7% at the third follow-up (p-value<0.05), and in Guatemala, where they increased from 16.2% at baseline to 20.4% at the second follow-up (p-value=0.06). Nevertheless, Guatemala continues to exhibit mostly home births.

## Quality scores over time for women who had an institutional delivery

Based on the WHO Positive Childbirth Experience guidance, the analysis reveals variations in the quality of intrapartum care across countries and follow-up periods. At Baseline, Nicaragua showed a higher quality of care in institutional deliveries, with 50.8% of births receiving the maximum score of 4. In contrast, Chiapas, Guatemala, and Honduras had lower quality scores, with only 18.3% of births in Chiapas, 23% in Guatemala and 21.3% Honduras achieving a score of 4.

A positive trend in adherence to quality domains was observed throughout the follow-up periods, particularly in Honduras, Nicaragua, and Chiapas, indicating improvements in intrapartum care of institutional deliveries over time.

This trend coincides in both time and location with the implementation of the SMI. Fig 3 illustrates the proportion of each quality domain score (ranging from 0 to 4) across countries and follow-up periods.

In areas where most pregnant women have institutional births, higher quality, as defined by WHO [5], correlates with a positive experience, encouraging the population to choose this option. In contrast, in Guatemala and Chiapas, where

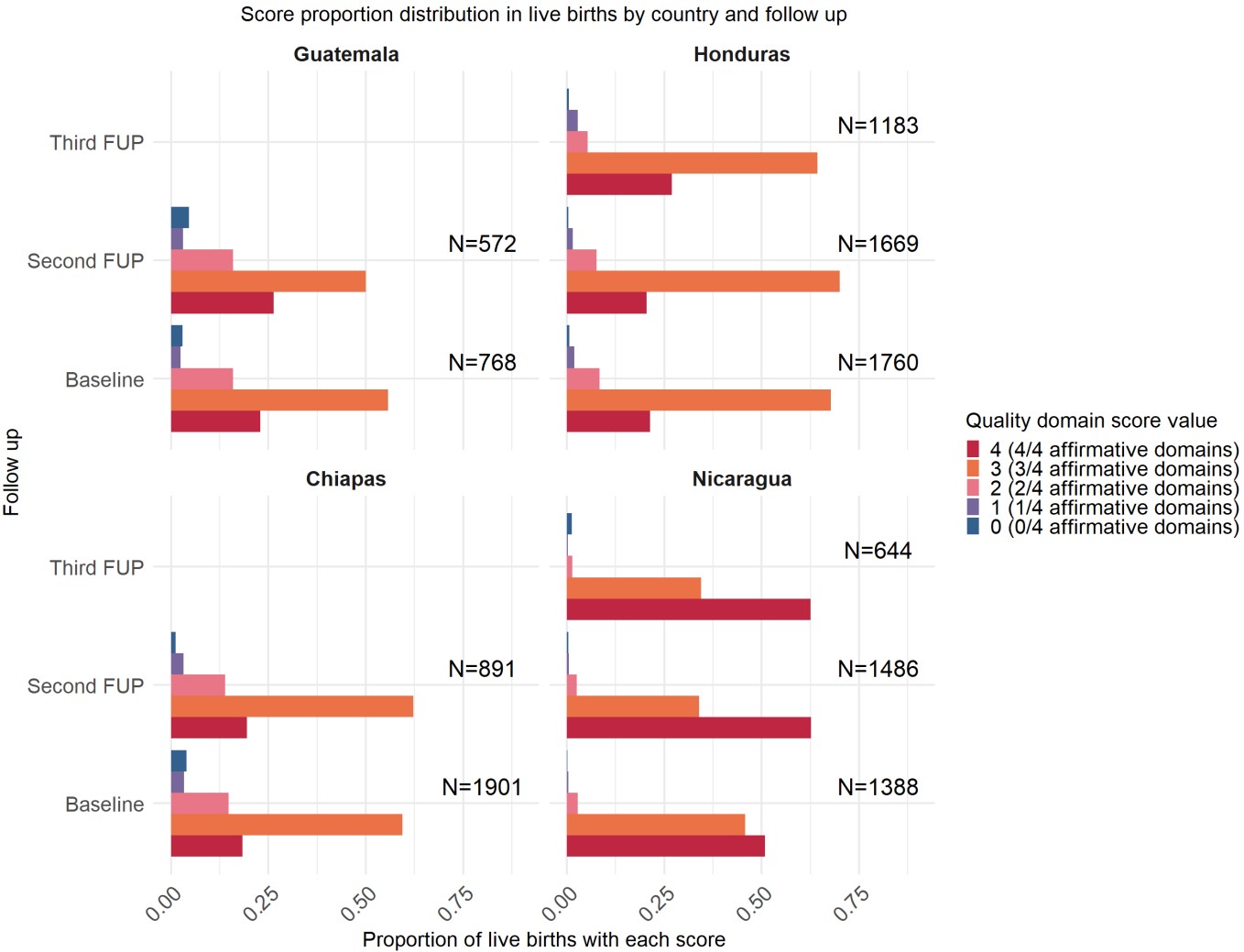

**Fig 3. Proportion of institutional live births in each domain (quality score ranging from 0 to 4) across different countries and follow-up periods (Baseline, Second FUP and Third FUP).** Guatemala and Chiapas did not have Third FUP surveys.

the percentage of institutional births is much lower, a reduced rate of positive experience is seen, presenting lower scores according to the index proposed.

In Honduras, no significant changes were observed in the "Respectful Treatment" domain. However, a significant improvement was found in "Effective Communication", increasing from 95% (IC95: 93%−96.3%) at Baseline to 98.2% (IC95: 97.2%−99.2%) at the Third follow-up (p < 0.05). The "Accompaniment" domain also showed a significant increase, rising from 21.7% (IC95: 18.7%−24.7%) at Baseline to 27.7% (IC95: 22.2%−33.2%) at the Third follow-up.

In Chiapas, Respectful Treatment significantly improved, increasing from 91.8% (IC95: 89.8%−93.7%) at Baseline to 95.3% (IC95: 93.7%−96.9%) at the second follow-up. Effective Communication also showed a significant increase, rising from 79.9% (IC95: 76.1%−83.6%) at Baseline to 87.5%(IC95: 84.7%−90.3%) at the Second follow-up (p < 0.05). In contrast, no significant changes were observed in Accompaniment or Recommended Practices.

In Nicaragua, the "Respectful Treatment" domain showed an overall difference, though the direct comparison between Baseline and the Third follow-up was not statistically significant. The score increased from 98.5% (IC95: 97.8%−99.2%) at baseline to 100% at 36 months but then dropped back to 97.3% (IC95: 94.9%−99.6%) at 54 months. No significant changes were observed in "Effective Communication". However, a significant improvement was found in Accompaniment, increasing from 53.6% (IC95: 50.4%−56.9%) at Baseline to 66.2% (IC95: 59.5%−72.9%) at the Third follow-up (p < 0.05). The Recommended Practices domain also showed significant changes, rising from 96.1% (IC95: 94.2%−98.1%) at Baseline to 98.4% (IC95: 97.4%−99.3%) at the Third follow-up.

In Guatemala, none of the domains showed statistically significant changes over time.

Fig 4 shows the distribution of positive answers for each of the 4 domains.

From our analysis of institutional birth experiences by domain, the most significant differences were observed in the Accompaniment domain. Furthermore, a trend toward an increase in institutional births in these communities was shown. In Honduras, home birth decreased from 27.73% in Baseline to 7.66% at the Third follow-up, according to the trend published previously [8].

### Reasons for home birth over time for women who had a home delivery

Women who had a home birth were asked about their reasons for not choosing to give birth in a health facility. Fig 5 illustrates the percentage of responses related to each domain across follow-ups and countries.

In Honduras and Nicaragua, the primary reasons for choosing home birth are predominantly linked to accessibility issues, with 53.9% of women in Honduras and 39.4% in Nicaragua mentioning this factor. In contrast, cultural preferences play a more significant role in Guatemala and Chiapas. In Guatemala, 14.8% of women cited accessibility as a reason for home birth, while 65.9% mentioned cultural factors. Similarly, in Chiapas, 15.6% of women attributed their choice to accessibility, whereas 65.9% cited cultural preferences.

In Nicaragua and Honduras, the home birth choice is linked mainly to accessibility and cultural preferences. In the case of Honduras, the percentage of institutional births in the baseline was high. However, it also had an improvement throughout the follow-up period, showing statistical differences between Baseline and Second FUP and between Second and Third FUP (72%, 86%, 92%).

When analyzing the reasons for choosing home birth, in Guatemala 14.8% of home births were attributed to accessibility issues at Baseline. While accessibility-related reasons remained relatively stable (14.8% at Baseline vs. 12.8% at Second FUP), a notable increase was observed in cultural preferences for home birth (65.9% at Baseline vs. 72.5% at Second FUP). As accessibility issues were improved, the cultural preferences appear to be the main reason for home birth, as can be seen in Fig 5 for Honduras in the Third FUP.

A similar trend was observed in Chiapas, where accessibility-related reasons declined from 15.6% at baseline to 10.1% at Second FUP, while cultural preferences increased slightly (75.9% at Baseline vs. 77.4% at Second FUP).

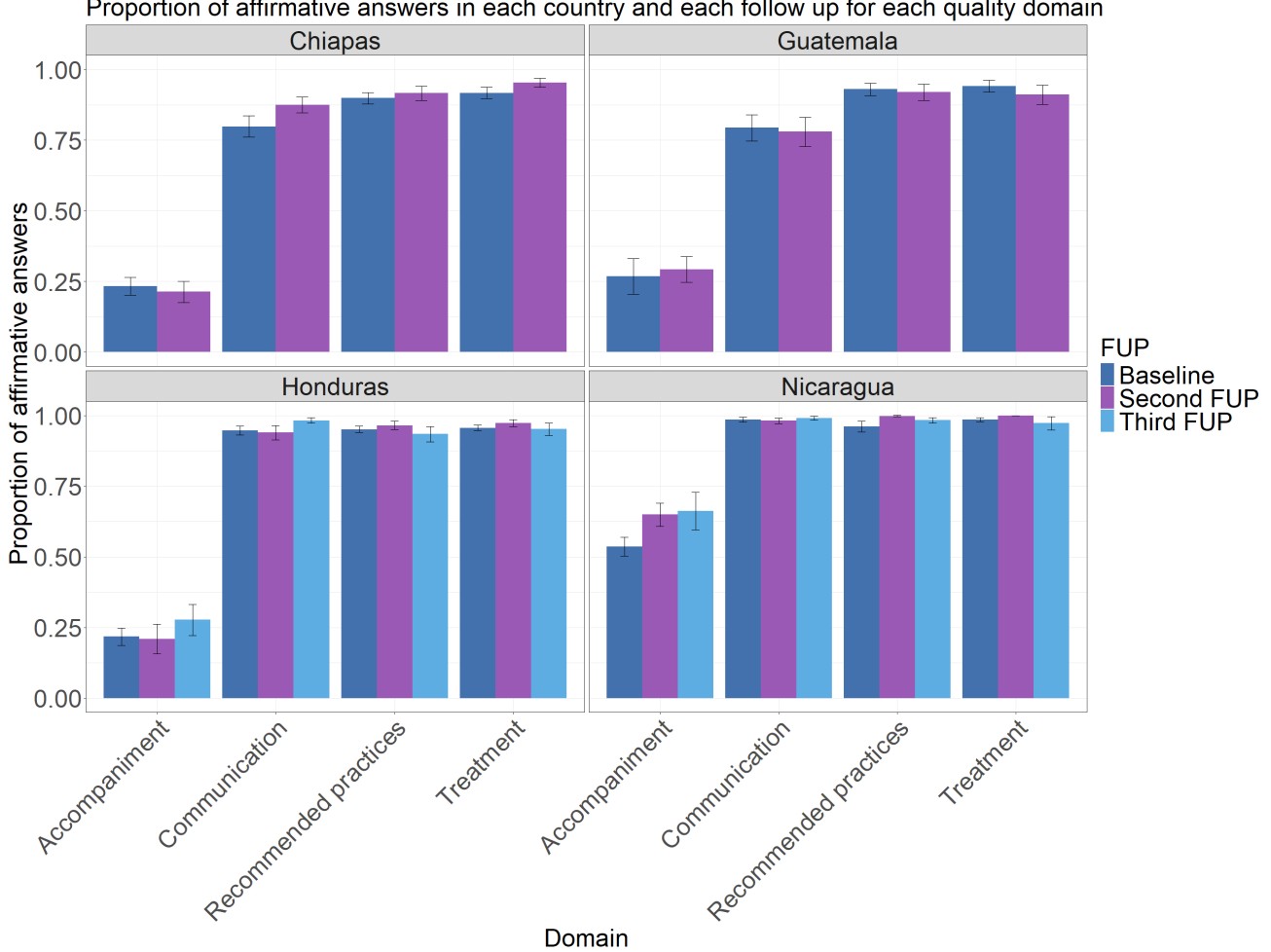

**Fig 4. Proportion of affirmative answers by country and follow-up for each quality domain.**

In Honduras, accessibility improved significantly, with the proportion of women choosing it as a reason for home birth decreasing from 53.9% at Baseline to 38.8% at Third FUP. This shift was accompanied by an increase in cultural preferences for home birth, rising from 30.5% at Baseline to 51.3% at Third FUP.

In Nicaragua, accessibility was cited as a reason for home birth by 39.4% of women at Baseline, increasing to 47.4% at the Third FUP. This change is likely related to the disruption of health services during the COVID-19 pandemic. In contrast, cultural preferences for home birth declined from 45.3% at baseline to 32.3% at Third FUP, suggesting that accessibility played a significant role in home birth decisions over time.

## Analysis of women who shifted their birth setting

To analyze women who shifted the birth delivery setting, those who had more than one child and changed their place of birth during the Second FUP were selected for this analysis.

From 10261 live births reported, 3251 were from women with more than 1 child, selecting the last 2 children for analysis (3096). Only those live births that represented a switch in the birth place (home-institutional or institutional-home) were

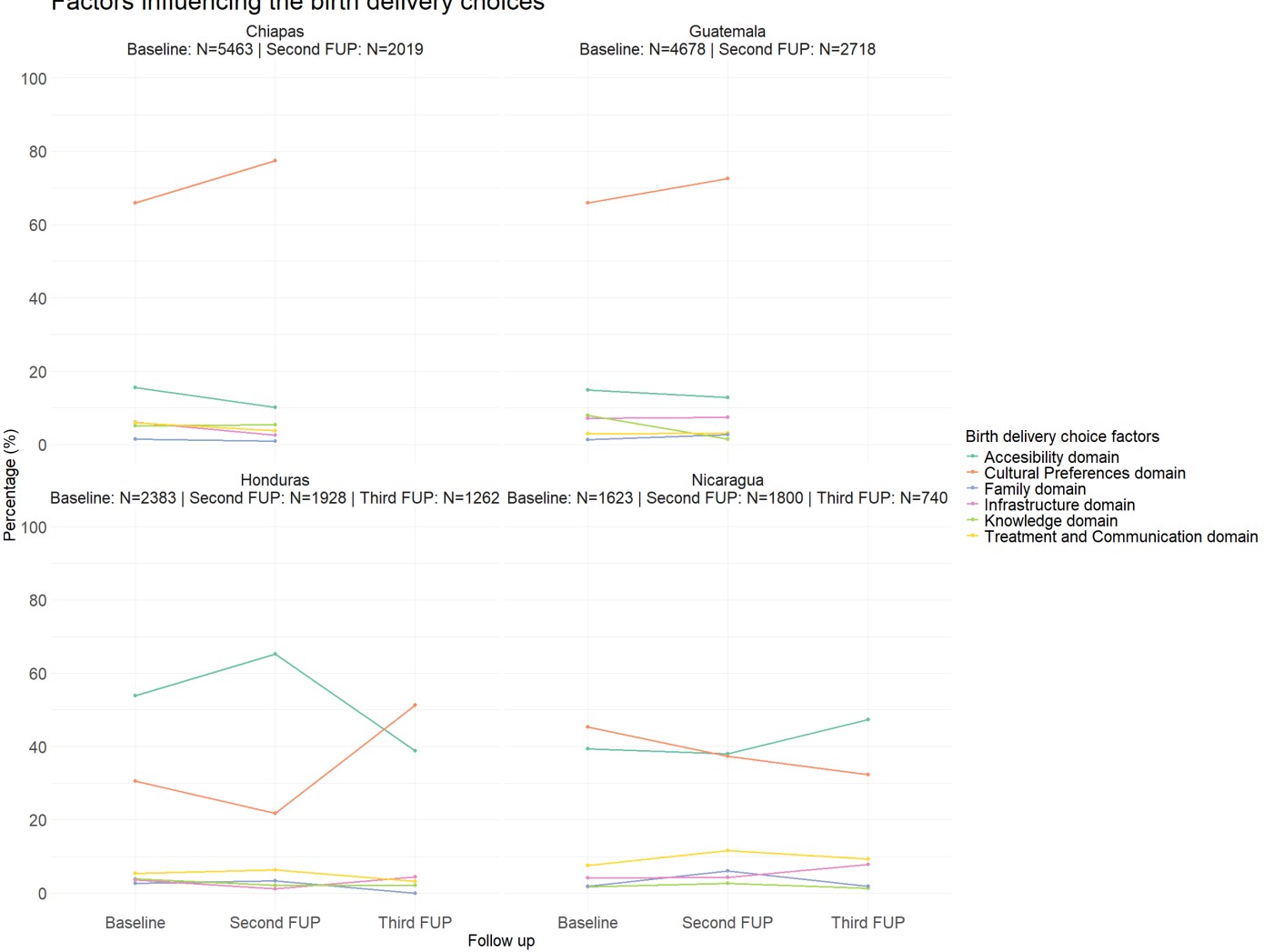

**Fig 5. Percentage of live births in home births with each domain by country and follow-up (Baseline, Second FUP, Third FUP).** Total number of live births in each follow-up is detailed on top **(N).**

analyzed, resulting in a total of 474 births (237 women), as shown in Fig 6. For each country, the distribution of women changing their place of birth is shown in Table 4.

### Analysis of women who shifted from institutional delivery to home birth

According to what was shown before in the main analysis, for the Second FUP in Chiapas, 62.2% of women reported a quality score of 3, and 19.5% an index of 4. The main reason for choosing a home birth was cultural preferences, followed by accessibility. Looking at this subgroup, there is a higher percentage of women reporting lower scores (33% reporting score between 0 and 2, compared to 18% in the general population) showing a worse quality index. The main reasons for home birth are the same.

In the case of the other three countries, the quality score reported was high, similar between this subgroup and the general population. As for the main reasons for choosing home birth, accessibility is the main reason in Nicaragua and

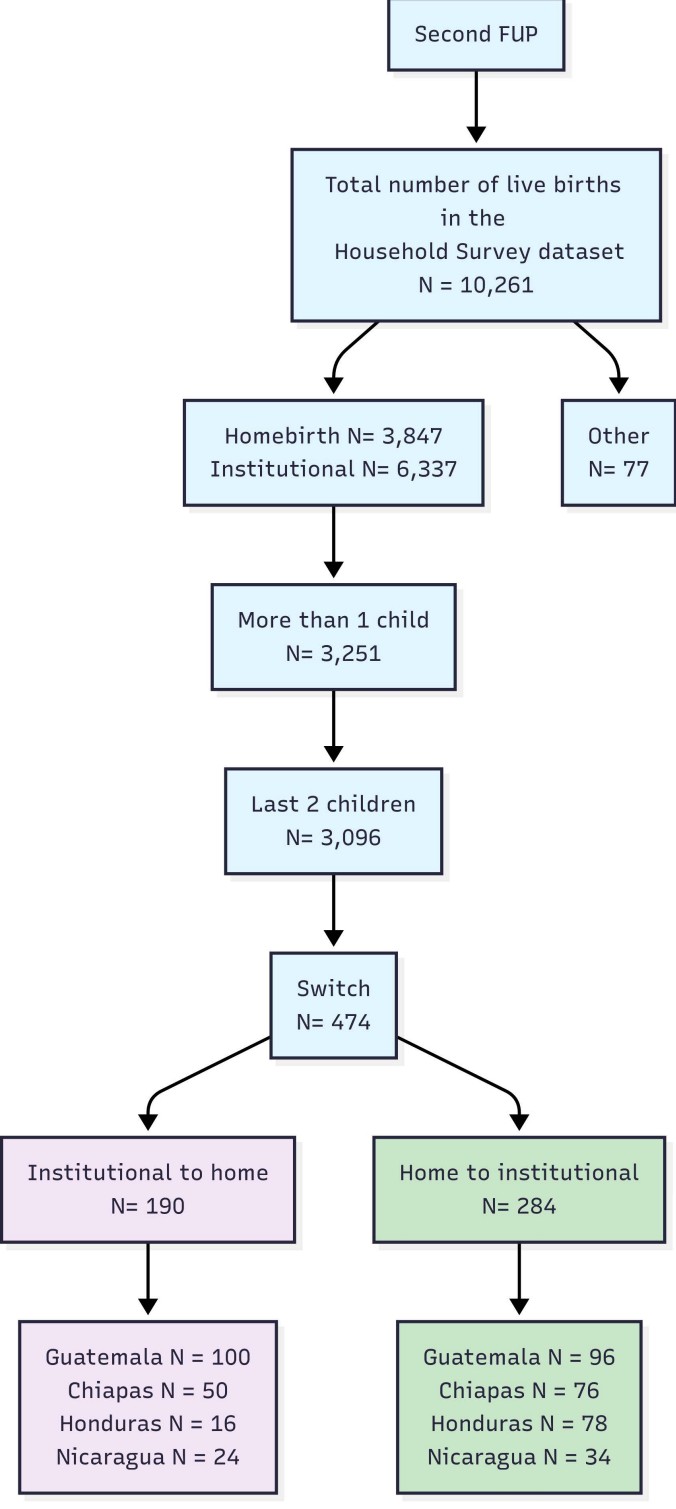

**Fig 6. Flowchart showing amount of live births analyzed from women who changed their place of birth during Second FUP.**

Honduras (56.2% and 100%, respectively), whereas the cultural preferences domain is the most chosen in Guatemala (48.8%), as was seen for the general population. This suggests that, although the perception of care was generally satisfactory, the decision to give birth at home was likely influenced by other factors.

All these findings are described in Fig 7, where the quality score for the first child is shown on the left and the reasons for home birth in the second child are shown on the right.

**Analysis of women who shifted from home to institutional delivery**

In this subgroup, the vast majority of women who gave birth at home did so primarily due to cultural preferences and accessibility challenges, as illustrated in Fig 8.

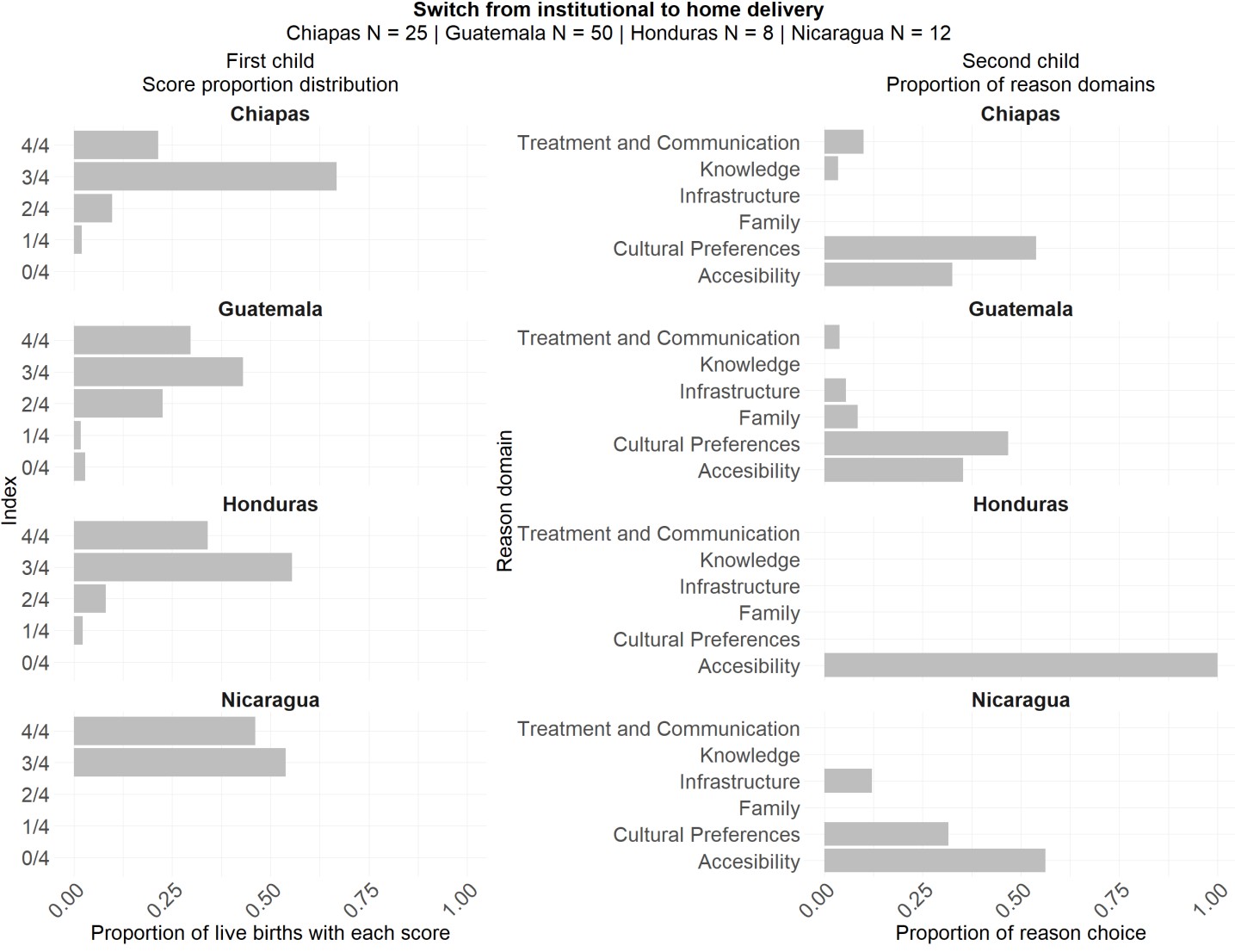

**Fig 7. Proportions of institutional birth scores imputed in first child and proportion of reason domains for second child among mothers who transitioned from institutional birth to home birth by country for the Second FUP.**

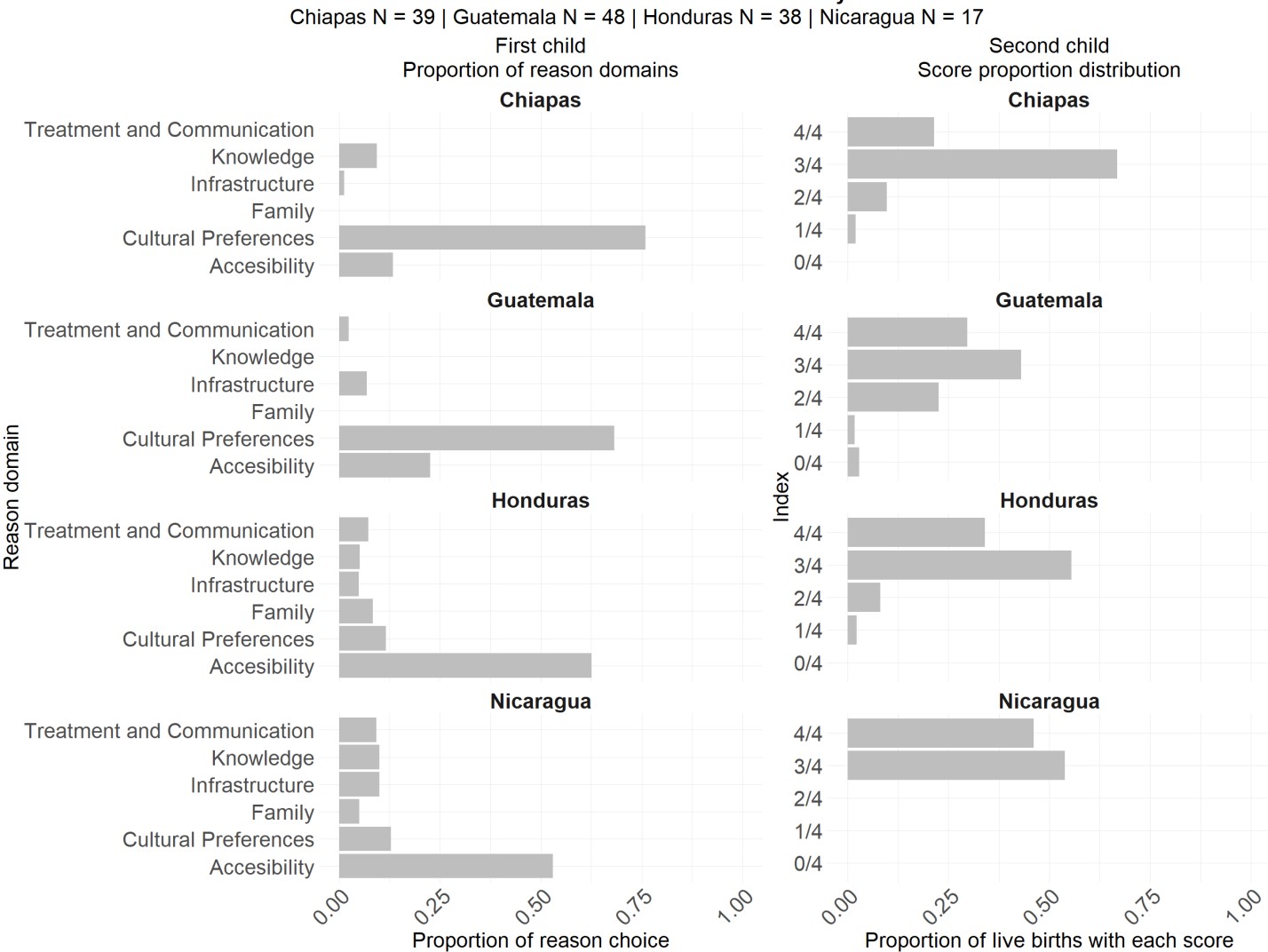

**Fig 8. Proportions of institutional birth scores imputed in second child and proportion of reason domains for first child among mothers who transitioned from home birth to institutional delivery by country for the Second FUP.**

In Guatemala, most women cited cultural preferences (68.1%) as the main reason for choosing home birth, while 22% pointed to accessibility issues. In Honduras, the pattern was reversed: accessibility was the predominant factor (62.6%), whereas only 11% of women referred to cultural preferences. In Chiapas, cultural preferences were the overwhelmingly dominant reason, reported by 75.9% of respondents. In Nicaragua, accessibility was again the main reason for home birth (53%), with cultural preferences cited by 12.8% of women. Nonetheless, there are other factors that increased, such as knowledge (9.4% in Chiapas and 10% in Nicaragua) and infrastructure (10% in Nicaragua) showing that when they had access to information and less infrastructure barriers, the institutional birth was more accepted.

However, when asked about their experience with institutional childbirth, the results indicated that care was generally suboptimal in Guatemala, Honduras, and Chiapas (Fig 8). In Guatemala, most institutional births were rated as Score 3 (43.99%), followed by Score 4 (29.74%), reflecting a moderate quality of care with clear room for improvement. In

Honduras, while Score 3 was also the most frequent rating (55.5%), a larger share of births achieved Score 4 (34.03%), suggesting a somewhat higher overall quality compared to Guatemala. In Chiapas, the majority of institutional births were concentrated in Score 3 (66.84%), with only 21.48% reaching Score 4, indicating lower adherence to high-quality standards. By contrast, Nicaragua stood out with a more favorable distribution: 53.89% of births were rated as Score 3, and 46.11% as Score 4, representing one of the strongest performances among the four areas and suggesting relatively higher satisfaction with institutional childbirth experiences.

## Discussion

The findings of this study provide critical insights into institutional birth patterns and healthcare access in Mesoamerica. There is a clear trend of increasing institutional births over time as it has been reported in previous publications from the Initiative [8]. The shift varies across different areas, with some showing a more pronounced increase than others. This trend aligns with existing literature [9–12]. The home birth rate decrease seen in Fig 2 may be associated with increased health personnel sensibilization and capacitation on cultural preferences, that relies on women better knowledge and accessibility. Regarding cultural factors impact, they may require more time to show an impact on the change in the birth delivery setting of the communities. The improvement in accessibility—particularly in Honduras —may have contributed to the reduction in home births. In Honduras, the significant increase in institutional births underscores the complex relationship between accessibility improvements and cultural preferences. While greater knowledge and access have played a role in increasing institutional deliveries, cultural factors remain a powerful determinant of birth delivery choice in the region [14]. Addressing these deeply ingrained cultural preferences requires tailored intervention strategies to support the transition from traditional birthing practices to institutional deliveries. Some culturally sensitive practices that could facilitate this transition include allowing women to perform their own hygiene procedures, providing quiet, dimly lit, and private spaces instead of bright and sterile delivery rooms, offering traditional foods instead of unfamiliar hospital meals, ensuring that healthcare professionals explain medical interventions in the patient's native language and being accompanied by a person of their choice including traditional midwives [14].

Previous studies on the Salud Mesoamerica Initiative in Guatemala, Chiapas, and Panama identified several factors associated with increased institutional deliveries among indigenous women. These factors include the number of antenatal care visits, counseling on the benefits of institutional births, and the presence of a community health worker during childbirth. Additionally, higher satisfaction with institutional deliveries was reported when healthcare providers communicate with users in their indigenous language, highlighting the importance of cultural and linguistic adaptations in medical settings. These findings were based on SMI baseline data [13].

No matter WHO has recognized the importance of allowing women to be accompanied by a person of their choice during childbirth [5] this preference became particularly challenging for the medical team not only during the COVID-19 pandemic, but also due to the lack of specific capacitation on communication with the family during emergency situation, potentially impacting women's healthcare decisions, as seen in Nicaragua.

In Chiapas, barriers to the availability of medical personnel and restrictive hours for health services have been published [15]. Additionally, women mentioned barriers to access (economic, geographic, linguistic, and cultural barriers) to health services, as well as invasive and offensive hospital practices performed by health system personnel, which hampered the quality of care they can provide. Traditional birth attendants participating in intercultural settings had expressed the lack of autonomy and exclusion they experienced as not being considered part of the care team. The users point to the importance of having their traditional birth attendants and families present during childbirth to support, empower, and help them in informed decision-making. It has also been reported the need to allow women to use their own clothing during birth; the preference of female staff and native language speaking has also been reported. Evidence from the study suggests the presence of important barriers to the utilization of institutional labor and delivery services in these communities in spite of the intercultural strategies implemented. It is essential to consider strengthening intercultural models of

care, to sensitize personnel towards cultural needs, beliefs, practices and preferences of indigenous women, with a focus on human rights, gender equity and quality of care.

Regarding the switch in the birth setting, not only cultural preferences play a significant role in giving birth at home, but also accessibility and knowledge. Our findings suggest that improving the quality of institutional childbirth care—particularly in terms of respectful treatment, effective communication, and responsiveness to cultural needs—could be key to increasing institutional births in these regions. Addressing these issues could help bridge the gap between accessibility and patient-centered care, making institutional births a more attractive and viable option for women who currently prefer home deliveries.

## Limitations

Initially, it should be considered that this study was performed with secondary data so we were not able to make any changes in the data collection survey. Also, it addresses different levels but in a fragmented way, so there is no key to match at individual level data between household and health facilities.

Some other limitations of our study should be highlighted, as the potential presence of courtesy bias [16]. In this regard, the near-universal indication by women that they would return to the same health centers, regardless of their previous experience, suggests a need for more nuanced data collection and interpretation methods [17,18]. Also, the third follow-up was conducted during the COVID-19 pandemic, a critical contextual factor that fundamentally altered the relationship between individuals and healthcare systems. This unprecedented global health crisis likely influenced women's healthcare-seeking behaviors and perceptions of medical institutions.

## Conclusion

The institutionalization of delivery has saved a significant number of lives by providing critical intervention during the most high-risk stage of pregnancy. However, the findings of this study demonstrate that increasing coverage and physical accessibility—as seen notably in Honduras—is only one facet of maternal health equity. The persistence of home births, particularly among indigenous and rural populations in Chiapas and Guatemala, highlight that clinical safety must be accompanied by cultural-based care to be sustainable.

Our analysis reveals that barriers such as the prohibition of birth companions (exacerbated during the COVID-19 pandemic), the exclusion of traditional birth attendants, and the lack of linguistic adaptations remain powerful discouraging causes. For the most vulnerable women, the decision to seek institutional care is a complex trade-off between the desire for a clinically secure environment and the need for an experience that respects their autonomy, clothing, and traditional practices.

To ensure long-term improvements in maternal and neonatal outcomes, regional health policies must fulfill the criteria of positive childbirth experience from OMS. Strengthening healthcare systems in Mesoamerica requires not only infrastructure but also the sensibilization of clinical personnel toward the human rights and sociocultural preferences of women. Ensuring a positive childbirth experience is not an alternative to medical safety, but is an essential bridge that will finally close the gap between vulnerable communities and life-saving institutional care.

## Supporting information

**S1 Table. Variables related to quality of care – analysing institutional birth.** List of the corresponding household survey questions related to the four domains in institutional birth.
(DOCX)

**S2 Table. Variables related to reasons for home birth- analysing home birth.** List of the corresponding household survey questions related to the six domains in home birth.
(DOCX)

## Author contributions

**Conceptualization:** Lucia Bartolomeu, Natalia Basualdo, Federico Rolón, Adolfo Rubinstein, Camila Volij, Matt Simon, Brittany Hagedorn, Diego Ríos-Zertuche, Santiago Esteban, Adrián Santoro.

**Data curation:** Lucia Bartolomeu, Adrián Santoro.

**Formal analysis:** Lucia Bartolomeu, Adrián Santoro.

**Methodology:** Lucia Bartolomeu, Natalia Basualdo, Federico Rolón, Adolfo Rubinstein, Brittany Hagedorn, Diego Ríos-Zertuche, Adrián Santoro.

**Project administration:** Adrián Santoro.

**Supervision:** Adolfo Rubinstein, Adrián Santoro.

**Visualization:** Lucia Bartolomeu.

**Writing – original draft:** Lucia Bartolomeu, Natalia Basualdo, Federico Rolón, Adolfo Rubinstein.

**Writing – review & editing:** Lucia Bartolomeu, Natalia Basualdo, Federico Rolón, Adolfo Rubinstein, Camila Volij, Matt Simon, Brittany Hagedorn, Diego Ríos-Zertuche, Santiago Esteban, Adrián Santoro.

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
