## [Decision Letter · Decision Letter 0]

7 Nov 2025

Dear Dr. Bartolomeu,

Thank you for submitting your manuscript to PLOS ONE. After careful consideration, we feel that it has merit but does not fully meet PLOS ONE’s publication criteria as it currently stands. Therefore, we invite you to submit a revised version of the manuscript that addresses the points raised during the review process.

We look forward to receiving your revised manuscript.

Kind regards,

Nicolas Padilla-Raygoza, Doctorate in Sciences

Academic Editor

PLOS ONE

Journal Requirements:

Additional Editor Comments (if provided):

The authors should take into accoun all comments from reviewers. They should re-submit the manuscript highlight all changes in the manuscript.

Reviewers' comments:

Reviewer's Responses to Questions

**Comments to the Author**

1. Is the manuscript technically sound, and do the data support the conclusions?

Reviewer #1: Partly

Reviewer #2: Partly

2. Has the statistical analysis been performed appropriately and rigorously?

Reviewer #1: Yes

Reviewer #2: I Don't Know

3. Have the authors made all data underlying the findings in their manuscript fully available?

Reviewer #1: Yes

Reviewer #2: No

4. Is the manuscript presented in an intelligible fashion and written in standard English?

Reviewer #1: Yes

Reviewer #2: Yes

Reviewer #1: The research is very timely because of the problems it aims to solve (high rates of maternal and infant mortality), and also because it is within the framework of an international program called "Mesoamerica Health Initiative", which is an "innovative development model that supports the transformation of national health systems to expand coverage, access and use of health services among the poorest women and children in Mesoamerica". It is a results-based financing model, a long-term effort focused on improving maternal and child health in the region, with funding from a public-private partnership. The initiative prioritized quality care, equity, and the use of evidence to guide policies and interventions targeting the most disadvantaged populations in participating countries.

Therefore, this research provides results that show the "fruits" of this funding aimed at improving maternal and child health care programs, if the Mesoamerica Health Initiative is based on results, with those of this study, it can make a partial evaluation of the improvement in the quality of care for mothers and their children. The summary is very well structured, but it would be better if it mentioned the objective and the materials and methods more specifically, as most of this section mentions the results. Background and justification are well founded.

Material and methods.

It is important to mention the time or period of data collection, review of files, the years that were consulted are shown, that is, in which years the women gave birth at home or in institution, but not how long it took to analyze the information, for example, from such a month to such a month and in what year. they only mention that the databases were consulted on April 1, 2024. But it wasn't in a single day that the information obtained was analyzed, or was it? It is important to mention this methodological information so that if someone wants to duplicate the study it is more appropriate and verify that the results are similar, to validate the information, because it is derived from an international program that is financed by public and private economic resources. They report that "The surveys employed a two-stage random sampling method to efficiently collect data and ensure that the results accurately reflected the general population." What did they randomize?

From the wording, it seems that the study was carried out directly with participants who were chosen at random. But in the next paragraph they mention that they used 90 data from household surveys carried out in several countries, and that these were from women aged 15 to 49 years, who had had a live birth in 5 years prior to the surveys. In other words, the study was not really observational, because the women were not followed, only the surveys about their births and the decision to have them at home or in institutions were consulted. Therefore, by using information from secondary sources, the study is considered retrospective. The questions of the surveys that focused on labor and delivery were analyzed, in three moments, that is, three obstetric events, if they had a cesarean section at the beginning they were not included in the present study, because automatically this is performed in an institution, but they do not mention if there was any case that had had labor at the beginning and in the following obstetric events ended in cesarean section, were they eliminated from the analysis?

This part of the domains is very interesting, but a doubt remains, why did they not ask the same questions in both cases, home and institutional births?, Why did they analyze with different domains, if it was a question of comparing both scenarios of the obstetric event? It is well known that the institution will have better infrastructure and care equipment, especially for medium to high-risk births for the mother and the newborn. There are domains that are compatible for both places: (home birth) respectful treatment, effective communication, accompaniment, best practices and (institutional birth), knowledge, accessibility, respectful treatment and effective communication, family, cultural preferences. Only the domain of infrastructure could be very different, but the other domains are comparable, since sometimes there is more respectful treatment when the birth is carried out at home than in the institution, or the family can participate in the care of the birth, which is not allowed in some institutions and the woman has her birth among unknown and strange people who are sometimes very technical. Communication in addition to being effective is usually affective when the birth is performed at home, and so there are some domains that could be better in home birth than in the institution. This study aims to highlight the importance of childbirth taking place in a health institution, so strategies must be designed and implemented aimed at promoting gentle and affective childbirth. And if the problem is that communities and developing countries don't have institutions or access to them, then midwives should be educated and their work supervised, because it is also well known that some midwives have empirical knowledge and skills to attend home birth, even in a more physiological way. In addition, the women know these midwives and trust their expertise.

Statistical analysis. Home births were compared with institutional births, for each domain? In the analysis of women who changed places, they report that "Within each category, we analyzed the reasons for choosing home birth and the quality of care perceived by health centers in the case of institutional births, as was done in the main analysis." But they do not put in themselves what were the reasons why they changed places, they only mention the domains. The changes in the place of delivery from house to institution and from institution to home are analyzed, and the fulfillment of the domains that were evaluated are mentioned, and the scores achieved in each domain, however a bias or tendency is observed to say that there is an increase in institutional births, referring that "A positive trend was observed in the fulfillment of the quality domains throughout the follow-up periods. indicating improvements in intrapartum care of institutional deliveries over time." However, no data is mentioned to support this fact. Because women's reasons for changing places to give birth are not placed.

It refers to the fact that "it correlates with a positive experience, which encourages the population to choose this option" but again no results are placed to support this fact, that is, what women think, for example, what their motives, thoughts or beliefs are for giving birth at home and then why they changed the institution, or because in the institution and then at home. They emphasize the changes of place from home to institution, but without placing the feelings of women.

It is important to emphasize that respectful treatment did not have significant differences, the following domains analyze them saying that, if there were differences between the domains and between the number of follow-up moments, but they do not say which was more positive, they should interpret for readers who are not experts, to highlight the importance of this research. They report that in Chiapas (the only specific place in the country Mexico, the others only mention the countries do not specify states, municipalities, districts), respectful treatment improved significantly, however, they do not mention where, in the institution or at home? Apparently, significant differences were observed in all domains, only that they do not mention in which of the two places, at home or in the institution, these results are biased in favor of the institutions, so that there would be no confusion they should have explained at the beginning that the comparison would be made between home birth and institution birth, that is, explain that the first results refer to which place (for example, home) and against which they are compared (for example, institution), and if they are doing it by follow-up time number, then they should also also place the place home or health institution.

According to the results presented, the conclusions are correctly supported, however, there is still a bias to favor institutionalized childbirth, despite the fact that in the discussion they refer to the importance of traditional midwives participating in births in rural communities, is it because of culture? Is there really better care in institutions?, the discussion places in a very good way that sometimes care by midwives empowers women in labor and allows the family to be present collaborating in the birth. It could improve the presentation of the results to be more in line with the discussion and improve the conclusion, there should be impartiality in the presentation of the results.

Reviewer #2: Introduction is not written in a way that establishes the reason for conducting the project. The readers need to know why you are encouraging women to have institutionalized birth. Do these countries have high maternal mortality statistics? Is there a problem with skilled birth attendant attending birth in home settings? All those reasons must be explained for the reader to understand why you are doing this project.

Methods need more details on how the data were analyzed.

I noticed that outcomes (healthy birth, no complications in mothers) were not mentioned in the list of factors that are important. Are these variables important to women? Need to comment on this in the introduction or methods. Conclusions sound a little bit like introduction, conclusion section needs a rewrite.

Discussion needs to be rewritten with more details, commenting on how outcomes play a role in this whole study and its future directions.

**Do you want your identity to be public for this peer review?** For information about this choice, including consent withdrawal, please see our Privacy Policy

Reviewer #1: No

Reviewer #2: No

---

## [Author Response · Author response to Decision Letter 1]

9 Mar 2026

Answers from the authors are in red.

5. Review Comments to the Author

Reviewer #1: The research is very timely because of the problems it aims to solve (high rates of maternal and infant mortality), and also because it is within the framework of an international program called "Mesoamerica Health Initiative", which is an "innovative development model that supports the transformation of national health systems to expand coverage, access and use of health services among the poorest women and children in Mesoamerica". It is a results-based financing model, a long-term effort focused on improving maternal and child health in the region, with funding from a public-private partnership. The initiative prioritized quality care, equity, and the use of evidence to guide policies and interventions targeting the most disadvantaged populations in participating countries.

Therefore, this research provides results that show the "fruits" of this funding aimed at improving maternal and child health care programs, if the Mesoamerica Health Initiative is based on results, with those of this study, it can make a partial evaluation of the improvement in the quality of care for mothers and their children. The summary is very well structured, but it would be better if it mentioned the objective and the materials and methods more specifically, as most of this section mentions the results. Background and justification are well founded.

We have reviewed the abstract to provide a more balanced overview, explicitly defining the primary objective of the study and material and methods, reducing the emphasis on results.

Material and methods.

It is important to mention the time or period of data collection, review of files, the years that were consulted are shown, that is, in which years the women gave birth at home or in institution, but not how long it took to analyze the information, for example, from such a month to such a month and in what year. they only mention that the databases were consulted on April 1, 2024. But it wasn't in a single day that the information obtained was analyzed, or was it? It is important to mention this methodological information so that if someone wants to duplicate the study it is more appropriate and verify that the results are similar, to validate the information, because it is derived from an international program that is financed by public and private economic resources. They report that "The surveys employed a two-stage random sampling method to efficiently collect data and ensure that the results accurately reflected the general population." What did they randomize?

Following these suggestions, we have clarified the distinction between the primary data collection period and our secondary analysis access date. The SMI Initiative conducted field operations, including household surveys and health facility reviews, between 2012 and 2021, while our study accessed the finalized, anonymized datasets on April 1st, 2024.

Additionally, we have expanded the description of the two-stage random sampling design to specify that randomization occurred first at the Primary Sampling Units (PSUs) level within target areas, and subsequently at the household/individual level within those PSUs.

From the wording, it seems that the study was carried out directly with participants who were chosen at random. But in the next paragraph they mention that they used 90 data from household surveys carried out in several countries, and that these were from women aged 15 to 49 years, who had had a live birth in 5 years prior to the surveys. In other words, the study was not really observational, because the women were not followed, only the surveys about their births and the decision to have them at home or in institutions were consulted. Therefore, by using information from secondary sources, the study is considered retrospective. The questions of the surveys that focused on labor and delivery were analyzed, in three moments, that is, three obstetric events, if they had a cesarean section at the beginning they were not included in the present study, because automatically this is performed in an institution, but they do not mention if there was any case that had had labor at the beginning and in the following obstetric events ended in cesarean section, were they eliminated from the analysis?

The household surveys conducted by the initiative differ for those who gave birth in a hospital versus those who gave birth at home, using that fact as the differentiating factor. Therefore, there is no detailed information about the birth history.

This part of the domains is very interesting, but a doubt remains, why did they not ask the same questions in both cases, home and institutional births?, Why did they analyze with different domains, if it was a question of comparing both scenarios of the obstetric event? It is well known that the institution will have better infrastructure and care equipment, especially for medium to high-risk births for the mother and the newborn. There are domains that are compatible for both places: (home birth) respectful treatment, effective communication, accompaniment, best practices and (institutional birth), knowledge, accessibility, respectful treatment and effective communication, family, cultural preferences. Only the domain of infrastructure could be very different, but the other domains are comparable, since sometimes there is more respectful treatment when the birth is carried out at home than in the institution, or the family can participate in the care of the birth, which is not allowed in some institutions and the woman has her birth among unknown and strange people who are sometimes very technical. Communication in addition to being effective is usually affective when the birth is performed at home, and so there are some domains that could be better in home birth than in the institution. This study aims to highlight the importance of childbirth taking place in a health institution, so strategies must be designed and implemented aimed at promoting gentle and affective childbirth. And if the problem is that communities and developing countries don't have institutions or access to them, then midwives should be educated and their work supervised, because it is also well known that some midwives have empirical knowledge and skills to attend home birth, even in a more physiological way. In addition, the women know these midwives and trust their expertise.

In this case we have the same limitation related to using a secondary source as in the point answered before. The household surveys differ for those who had an institutional birth than those giving birth at home. In case of the institutional deliveries, they developed questions related to quality. In case of home deliveries, the questions were related to reasons for giving birth at home.

Statistical analysis. Home births were compared with institutional births, for each domain? In the analysis of women who changed places, they report that "Within each category, we analyzed the reasons for choosing home birth and the quality of care perceived by health centers in the case of institutional births, as was done in the main analysis." But they do not put in themselves what were the reasons why they changed places, they only mention the domains. The changes in the place of delivery from house to institution and from institution to home are analyzed, and the fulfillment of the domains that were evaluated are mentioned, and the scores achieved in each domain, however a bias or tendency is observed to say that there is an increase in institutional births, referring that "A positive trend was observed in the fulfillment of the quality domains throughout the follow-up periods. indicating improvements in intrapartum care of institutional deliveries over time." However, no data is mentioned to support this fact. Because women's reasons for changing places to give birth are not placed.

This section of the work is related to a specific sub-analysis performed for the second follow-up, exploring setting transitions for women with multiple pregnancies. In case of women who had an institutional birth, household survey available data was related to the quality of care and the questions aligned with the WHO framework for a positive childbirth experience were selected. For women having a home birth available data was based on the reasons for their home-birth decision.

It refers to the fact that "it correlates with a positive experience, which encourages the population to choose this option" but again no results are placed to support this fact, that is, what women think, for example, what their motives, thoughts or beliefs are for giving birth at home and then why they changed the institution, or because in the institution and then at home. They emphasize the changes of place from home to institution, but without placing the feelings of women.

It is important to emphasize that respectful treatment did not have significant differences, the following domains analyze them saying that, if there were differences between the domains and between the number of follow-up moments, but they do not say which was more positive, they should interpret for readers who are not experts, to highlight the importance of this research. They report that in Chiapas (the only specific place in the country Mexico, the others only mention the countries do not specify states, municipalities, districts), respectful treatment improved significantly, however, they do not mention where, in the institution or at home? Apparently, significant differences were observed in all domains, only that they do not mention in which of the two places, at home or in the institution, these results are biased in favor of the institutions, so that there would be no confusion they should have explained at the beginning that the comparison would be made between home birth and institution birth, that is, explain that the first results refer to which place (for example, home) and against which they are compared (for example, institution), and if they are doing it by follow-up time number, then they should also also place the place home or health institution.

According to the results presented, the conclusions are correctly supported, however, there is still a bias to favor institutionalized childbirth, despite the fact that in the discussion they refer to the importance of traditional midwives participating in births in rural communities, is it because of culture? Is there really better care in institutions?, the discussion places in a very good way that sometimes care by midwives empowers women in labor and allows the family to be present collaborating in the birth. It could improve the presentation of the results to be more in line with the discussion and improve the conclusion, there should be impartiality in the presentation of the results.

In this case we have the same limitation related to using a secondary source as in the point answered before. The household surveys differ for those who had an institutional birth than those giving birth at home. In case of the institutional deliveries, they developed questions related to quality. In case of home deliveries, the questions were related to reasons for giving birth at home. There were no questions related to midwives during birth labor in this section.

Reviewer #2: Introduction is not written in a way that establishes the reason for conducting the project. The readers need to know why you are encouraging women to have institutionalized birth. Do these countries have high maternal mortality statistics? Is there a problem with skilled birth attendant attending birth in home settings? All those reasons must be explained for the reader to understand why you are doing this project.

We have incorporated data from the SMI Initiative to justify the project's focus and we have clarified the "why" by highlighting life-threatening complications.

Methods need more details on how the data were analyzed.

We have significantly expanded the "Data Analysis" subsection to provide a comprehensive description of the analytical procedures.

I noticed that outcomes (healthy birth, no complications in mothers) were not mentioned in the list of factors that are important. Are these variables important to women? Need to comment on this in the introduction or methods. Conclusions sound a little bit like introduction, conclusion section needs a rewrite.

In this case we have the same limitation related to using a secondary source as in the point answered before. The household surveys differ for those who had an institutional birth than those giving birth at home. In case of the institutional deliveries, they developed questions related to quality. In case of home deliveries, the questions were related to reasons for giving birth at home. There were no questions related to complications in this section, that information was in the facility form that was not possible to match with household. We have rewritten conclusions and added a section of limitations.

Discussion needs to be rewritten with more details, commenting on how outcomes play a role in this whole study and its future directions.

We have reordered the Discussion, Conclusions and Limitations sections.

---

## [Editor Report · Decision Letter 1]

11 Mar 2026

Examining women's choice between home and institutional births: insights from the Salud Mesoamérica Initiative (SMI)

PONE-D-25-43963R1

Dear Dr. Bartolomeu,

We’re pleased to inform you that your manuscript has been judged scientifically suitable for publication and will be formally accepted for publication once it meets all outstanding technical requirements.

Kind regards,

Nicolas Padilla-Raygoza, Doctorate in Sciences

Academic Editor

PLOS One

Additional Editor Comments (optional):

All comments were answered and taken into account in the manuscript, the article is accept.
---

## [Editor Report · Acceptance letter]

PONE-D-25-43963R1

PLOS One

Dear Dr. Bartolomeu,

I'm pleased to inform you that your manuscript has been deemed suitable for publication in PLOS One. Congratulations! Your manuscript is now being handed over to our production team.

Kind regards,

on behalf of

Dr. Nicolas Padilla-Raygoza

Academic Editor

PLOS One